# An open-label pilot study of recombinant granulocyte-colony stimulating factor in Friedreich's ataxia

Kevin C. Kemp [1✉], Anastasia Georgievskaya[1], Kelly Hares[1], Juliana Redondo[1], Steven Bailey[1], Claire M. Rice[1], Neil J. Scolding[1], Chris Metcalfe[2] & Alastair Wilkins[1]

Friedreich's ataxia (FA) is an inherited progressive neurodegenerative disease for which there is no proven disease-modifying treatment. Here we perform an open-label, pilot study of recombinant human granulocyte-colony stimulating factor (G-CSF) administration in seven people with FA (EudraCT: 2017-003084-34); each participant receiving a single course of G-CSF (Lenograstim; 1.28 million units per kg per day for 5 days). The primary outcome is peripheral blood mononuclear cell frataxin levels over a 19-day period. The secondary outcomes include safety, haematopoietic stem cell (HSC) mobilisation, antioxidant levels and mitochondrial enzyme activity. The trial meets pre-specified endpoints. We show that administration of G-CSF to people with FA is safe. Mobilisation of HSCs in response to G-CSF is comparable to that of healthy individuals. Notably, sustained increases in cellular frataxin concentrations and raised PGC-1α and Nrf2 expression are detected. Our findings show potential for G-CSF therapy to have a clinical impact in people with FA.

[1] Translational Health Sciences, Bristol Medical School, University of Bristol, Southmead Hospital, Bristol, UK. [2] Population Health Sciences, Bristol Medical School, University of Bristol, Canynge Hall, Bristol BS8 2PR, UK. ✉email: kevin.kemp@bristol.ac.uk

Friedreich's ataxia (FA) is a progressive neurodegenerative disease for which there is no proven disease-modifying treatment. It is the commonest hereditary ataxia, affecting at least 1 in every 50,000 people worldwide. Symptoms generally begin in childhood and affected individuals experience progressive accumulation of neurological disability including limb and trunk ataxia, dysarthria, sensory neuronopathy and pyramidal weakness[1]. In over 95% of cases, the disease is caused by a homozygous GAA.TCC trinucleotide repeat expansion mutation within intron 1 of the *FXN* gene, leading to transcriptional repression of the mitochondrial protein frataxin[2]. Frataxin has multiple functions including formation of iron-sulphur clusters, mitochondrial homoeostasis and antioxidant protection[3]. Within the nervous system frataxin deficiency is associated with degeneration of the cerebellum, including the dentate nucleus, dorsal root ganglion neurons, peripheral sensory nerves and the spinal cord[1].

Since the discovery of the genetic basis for FA there have been significant improvements in our understanding of how frataxin deficiency causes disease. Despite this, progress to identify an effective treatment to limit disease progression and reduce long-term disability has been limited; only symptomatic treatments of modest efficacy are currently available, and none reverse damage or restore neurological function. Disease-modifying therapy for people with FA therefore represents a major unmet clinical need.

A variety of different therapeutic strategies for FA have been explored in clinical and preclinical studies. Pharmacological agents have been trialled with the aim of attenuating the downstream effects of frataxin deficiency, namely oxidative stress, iron overload and mitochondrial dysfunction. These agents showed promise in preclinical and early phase studies, but subsequent randomised controlled trials of such drugs including idebenone, deferiprone, or a combination of coenzyme Q10 and vitamin E failed to demonstrate a reduction in disease progression[4–6]. Agents including erythropoietin and interferon-gamma have been tested for their potential to upregulate frataxin with negative outcomes[7,8]. Histone deacetylase inhibitors, with the aim of reversing *FXN* silencing to increase frataxin transcription, have also been examined and early clinical trials of drugs such as nicotinamide have demonstrated evidence of promise in short-term studies[9].

Continued research and technological advances in regenerative medicine have increased interest in developing neuro-reparative stem cell-based therapies for people with FA. Using a humanised murine model of FA we demonstrated that treatment with the stem cell mobilising cytokines granulocyte-colony stimulating factor (G-CSF) and stem cell factor had pronounced effects on frataxin levels within the nervous system and improved clinical, biochemical and pathological parameters associated with the disease[10,11]. Moreover, we have also shown that these agents had neuroprotective effects in FA, increased stem cell mobilisation to sites of pathology and stimulate neural repair[10].

G-CSF is an approved drug widely used medically to treat neutropenia or stimulate the release of haematopoietic stem and progenitor cells from the bone marrow into the peripheral circulation prior to a peripheral blood (PB) stem cell harvest[12]. Consequently, G-CSF has a well-established safety record in clinical practice and the pharmacokinetics of G-CSF administration have been extensively studied. Together with preclinical evidence supporting a likely beneficial effect and the current lack of effective treatments for FA, the rationale for the clinical trial of disease modification using G-CSF was clearly established.

Here, we report the findings of an open-label pilot study of G-CSF administration in seven people with FA. We demonstrate that administration of G-CSF to people with FA is safe and tolerable, leads to effective haematopoietic stem cell (HSC) mobilisation and improves the biochemical profile of molecules relevant to the pathogenesis of FA including increasing frataxin levels. Our findings provide evidence to support an efficacy study of repeated courses of G-CSF in people with FA.

## Results

**Safety and tolerability outcomes**. Seven participants were recruited for the study between June 2018 and October 2018 (Fig. 1). Table 1 summarises participant demographics. Six participants completed the study; one withdrew following the first dose of G-CSF (day 1) due to nausea (participant no. 7; Table 1). Of the participants completing the study, three reported a single, mild adverse event during the period of G-CSF administration; musculoskeletal pain (predominantly in the legs) ($n = 2$) and mild headache ($n = 1$), all of which were listed in the trial protocol as known side effects and all resolved immediately after cessation of therapy. There were no serious adverse events. There was no evidence of changes in pulse rate, blood pressure (systolic and diastolic) or body temperature throughout the study (Table 2). Although normal pre-treatment, alkaline phosphatase levels were elevated and outside normal physiological ranges following G-CSF administration as expected (day 6; $P = 0.031$)[13], but these were not associated with symptoms and resolved spontaneously by day 19. All other parameters were within normal physiological ranges (Table 2).

**Haematological response**. The full blood count (FBC) on day 6 showed an appropriately marked rise (approximately 7-fold) in the total circulating number of white blood cells in response to G-CSF administration, corresponding with significantly elevated numbers of circulating neutrophils, lymphocytes, monocytes and eosinophils ($P = 0.031$) (Table 2). Mobilisation of circulating PB HSCs was also achieved in all participants. CD34+ cells increased from $7.5 \times 10^6$ (SEM ± 1.8) cells per l at baseline to $98.4 \times 10^6$ (SEM ± 21.6) following G-CSF administration; CD34+ CD133+ cells rose from $0.9 \times 10^6$ (SEM ± 0.4) cells per l to $23.8 \times 10^6$ (SEM ± 10.2) (day 5; both $P = 0.031$) (Fig. 2).

**Frataxin expression**. Mean baseline frataxin protein levels in isolated PB mononuclear cells (MNCs) and platelets were 2.43 pg per µg of total protein (range 0.54–3.72) and 1.60 pg per µg (range 0.75–2.56) respectively (Fig. 3a, b). Prolonged increases in

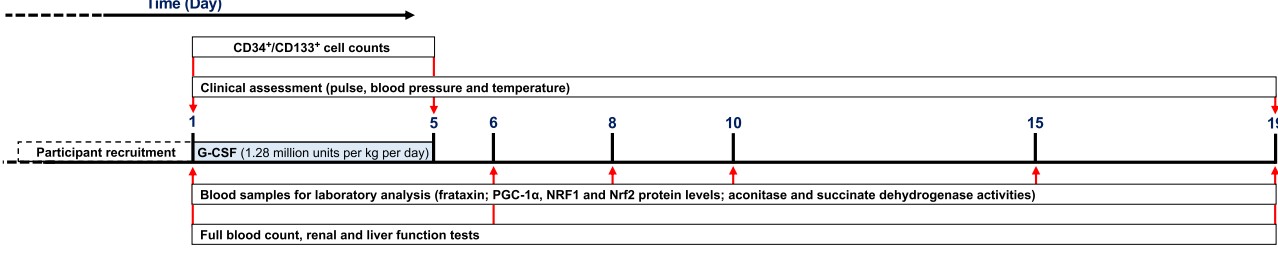

**Fig. 1 Trial timeline.** Red arrows indicate the day of intervention.

**Table 1 Baseline participant demographics.**

| Participant | Gender (M/F) | Age (years) | Age at symptom onset (years) | Age at diagnosis (years) | FDS | Speech score from ADL of FARS |
|---|---|---|---|---|---|---|
| 1 | M | 28 | 19 | 21 | 2.5 | 2 |
| 2 | F | 55 | 45 | 50 | 4.0 | 2 |
| 3 | F | 32 | 20 | 30 | 5.0 | 3 |
| 4 | M | 22 | 2 | 4 | 5.5 | 4 |
| 5 | F | 24 | 10 | 11 | 4.5 | 3 |
| 6 | F | 60 | 20 | 31 | 4.0 | 3 |
| 7[a] | M | 27 | 16 | 23 | 3.0 | 2 |

Friedreich's Ataxia Rating Scale (FARS) disability staging (FDS) using an ordinal score from 1 to 6 (no, minimal, mild, moderate, severe and total dependency)[33]. Speech score of activities of daily living (ADL) from FARS using an ordinal score from 0 to 4 (no, mildly affected, moderately affected, severely affected and unintelligible).
[a]Failed to complete study.

**Table 2 Changes in vital signs and selected laboratory values in response to G-CSF.**

| Test | Normal range | Day 1 (pre-administration) | Day 5/6 (post-administration) | Day 19 (follow-up) |
|---|---|---|---|---|
| Clinical vital signs | | | | |
| Temperature (°C) | | 37.37 (0.41) | 37.48 (0.38) | 37.48 (0.37) |
| Pulse rate (BPM) | | 83.17 (18.19) | 80.50 (13.10) | 86.33 (20.70) |
| Blood pressure (mmHg) | | | | |
| Systolic | | 131.50 (15.20) | 122.50 (17.07) | 132.17 (19.14) |
| Diastolic | | 78.33 (8.12) | 68.83 (16.52) | 74.50 (14.10) |
| Liver function | | | | |
| ALT (U/l) | 10–60 | 22.83 (5.46) | 30.33 (13.52) | 27.67 (12.96) |
| Albumin (g/l) | 35–50 | 41.83 (2.48) | 41.83 (1.94) | 41.67 (2.73) |
| Total bilirubin (μmol/l) | <21 | 7.67 (1.51) | 5.17 (0.41) | 11.00 (1.90)* |
| ALP (U/l) | 30–130 | 76.33 (20.32) | 222.17 (44.4)* | 87 (26.70) |
| Renal function | | | | |
| Sodium (mmol/l) | 133–146 | 141.00 (1.67) | 143.83 (1.17)* | 142.67 (1.86) |
| Potassium (mmol/l) | 3.50–5.30 | 4.32 (0.43) | 4.20 (0.28) | 4.56 (0.37) |
| Urea (mmol/l) | 2.50–7.80 | 4.40 (0.59) | 3.47 (1.23) | 4.30 (1.24) |
| Bicarbonate (mmol/l) | 22–29 | 25.60 (2.07) | 25.83 (2.23) | 26.20 (3.27) |
| Creatinine (μmol/l) | 45–104 | 69.33 (13.65) | 71.67 (10.95) | 62.33 (11.57) |
| Calcium (mmol/l) | 2.20–2.60 | 2.44 (0.03) | 2.55 (0.05) | 2.47 (0.06) |
| Adjusted calcium (mmol/l) | 2.20–2.60 | 2.44 (0.02) | 2.54 (0.07) | 2.47 (0.04) |
| Phosphate (mmol/l) | 0.80–1.50 | 1.10 (0.13) | 1.13 (0.22) | 1.05 (0.18) |
| Total protein (g/l) | 60–80 | 73.00 (2.12) | 74.50 (2.81) | 71.80 (2.59) |
| Urate (μmol/l) | 140–430 | 261.20 (64.43) | 406.33 (97.56) | 252.20 (56.77) |
| Full blood count | | | | |
| RBC ($10^{12}$/l) | 3.80–6.00 | 4.86 (0.51) | 4.77 (0.57) | 4.43 (0.42)* |
| MCV (fl) | 83–100 | 90.70 (3.82) | 91.63 (3.63) | 94.45 (6.70) |
| MCH (pg) | 27.0–32.0 | 29.92 (0.91) | 29.57 (1.15) | 30.05 (1.03) |
| MCHC (g/l) | 310–350 | 330.00 (4.43) | 322.83 (5.88) | 319.33 (21.65) |
| Platelets ($10^9$/l) | 150–450 | 284.83 (36.75) | 267.00 (18.25) | 363.17 (38.47)* |
| Haemoglobin (g/l) | 120–170 | 145.17 (12.78) | 140.67 (12.50) | 133.00 (10.10)* |
| White cell count ($10^9$/l) | 4.0–11.0 | 7.49 (2.40) | 51.81 (12.47)* | 5.58 (1.67) |
| Haematocrit (l/l) | 0.37–0.52 | 0.44 (0.04) | 0.44 (0.04) | 0.42 (0.06) |
| RDW (%) | 11.5–15.5 | 12.87 (0.63) | 13.57 (0.65)* | 13.68 (0.83)* |
| Neutrophils ($10^9$/l) | 1.50–8.00 | 4.88 (1.76) | 41.03 (7.89)* | 3.48 (1.49) |
| Lymphocytes ($10^9$/l) | 1.00–4.0 | 1.88 (0.58) | 5.23 (2.26)* | 1.49 (0.52) |
| Monocytes ($10^9$/l) | 0.2–1.0 | 0.55 (0.16) | 3.68 (0.97)* | 0.46 (0.09) |
| Eosinophils ($10^9$/l) | 0.0–0.5 | 0.13 (0.08) | 0.53 (0.41)* | 0.10 (0.07) |
| Basophils ($10^9$/l) | 0.0–0.2 | 0.04 (0.02) | 0.05 (0.05) | 0.05 (0.03) |

Mean (SD). Statistical comparisons are vs day 1 (baseline) using the sign test. Bicarbonate; calcium; adjusted calcium; phosphate; total protein; urate $n = 5$ biologically independent samples. All other tests $n = 6$ biologically independent samples. Clinical vital signs were performed on days 1, 5 and 19. All other tests were performed on days 1, 6 and 19. Source data are provided as a Source Data file.
ALT alanine aminotransferase, ALP alkaline phosphatase, RBC red blood cell, MCV mean corpuscular volume, MCH mean corpuscular haemoglobin, MCHC mean corpuscular haemoglobin concentration, RDW red cell distribution width.
*$P = 0.031$ (two-sided).

frataxin expression were evident following G-CSF administration in both MNCs and platelets (Fig. 3). Maximal 2.11 and 1.94 mean-fold increases in frataxin expression were observed in MNCs (day 10) and platelets (day 8), respectively, with the lower limit of the standard errors remaining above the null value of 1.0 at all time points (Fig. 3c, e). Pmax values for changes in frataxin expression were positive for all six participants over the duration of the study (Pmax ≠ 1.00; $P = 0.031$. Fig. 3d, f).

**Aconitase and succinate dehydrogenase activity.** In response to G-CSF, the mean-fold change in enzyme activity of aconitase and succinate dehydrogenase (SDH) rose steadily over the study

period with maximal 3.70 (day 15) and 3.04 (day 19) mean-fold changes respectively (Fig. 4a, c). For aconitase, the lower limit of standard error for changes in enzyme activity remained above the

null value of 1.0 for the entirety of the study (Fig. 4a). $P$max values for changes in aconitase activity were positive in four of the six participants ($P$max ≠ 1.00; $P$ = 0.69. Fig. 4b). $P$max values for changes in SDH activity were positive in five of the six participants; however, statistically, there was no evidence of changes SDH activity for the study duration ($P$max ≠ 1.00; $P$ = 0.22. Fig. 4d).

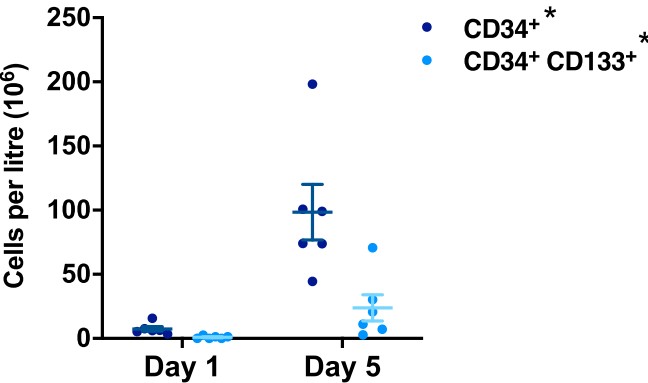

**Fig. 2 G-CSF induces mobilisation of HSCs into the peripheral blood.** The frequency of circulating CD34$^+$ and CD34$^+$ CD133$^+$ HSCs in the peripheral blood of participants with FA at baseline (day 1) or in response to G-CSF administration (day 5). Statistical comparisons are day 5 vs day 1 (baseline) using the sign test. *$P$ = 0.031 (two-sided). Graph depicts mean ± standard error; $n$ = 6 biologically independent samples. Source data are provided as a Source Data file.

**FA-associated proteins (PGC-1α, NRF1 and Nrf2).** Reduced activities of cell regulatory proteins PGC-1α, NRF1 and Nrf2 are associated with frataxin deficiency[14–16]. The expression of PGC-1α, NRF1 and Nrf2 proteins in isolated PB MNCs was measured in response to G-CSF administration (Fig. 5). Marked, rapid and sustained increases in PGC-1α expression were observed post G-CSF treatment, with a maximal 2.94 mean-fold increase in expression at day 6 (Fig. 5a). Peak Nrf2 expression followed at day 15 with a maximal 1.98 mean-fold increase (Fig. 5e). For both PGC-1α and Nrf2, the lower limit of standard error for changes in protein expression remained above the null value of 1.0 for the entirety of the study (Fig. 5a, e). For changes in both PGC-1α and Nrf2 expression, all 6 participants displayed positive $P$max values ($P$max ≠ 1.00; $P$ = 0.031. Fig. 5b, f). NRF1 displayed greater heterogeneity in response. There was no evidence of changes in NRF1 expression for the study duration (Fig. 5c); a positive $P$max

**Fig. 3 G-CSF administration increases frataxin protein levels in both PB MNCs and platelets.** The concentration of frataxin protein in (**a**) PB MNCs and (**b**) platelets over the 19-day study period from individual participants. The proportional changes in frataxin protein concentration in (**c**) PB MNCs and (**e**) platelets, relative to baseline (day 1) levels, over the 19-day study period (graph depicts mean ± standard error; $n$ = 6 biologically independent samples); data from individual participants are shown in graphs (**d**; MNCs) and (**f**; platelets). The sign test (two-sided) was used to compare $P$max values to a theoretical median of 1.0. Source data are provided as a Source Data file.

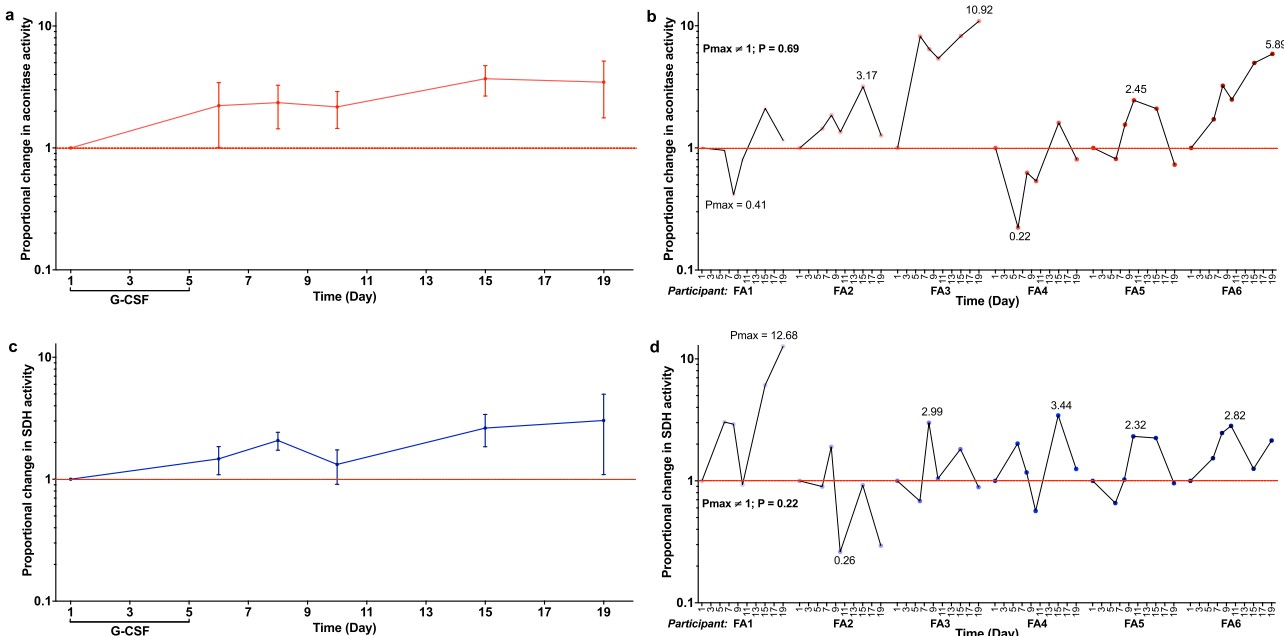

**Fig. 4 The activities of aconitase and SDH in peripheral MNCs post G-CSF administration.** The proportional changes in (**a**) aconitase and (**c**) SDH activity in PB MNCs, relative to baseline (day 1) levels, over the 19-day study period (graphs depict mean ± standard error; $n = 6$ biologically independent samples); data from individual participants are shown in graphs (**b**; aconitase) and (**d**; SDH). The sign test (two-sided) was used to compare $P$max values to a theoretical median of 1.0. Source data are provided as a Source Data file.

value was seen in three participants ($P$max $\neq 1.00$; $P = 1.00$. Fig. 5d).

## Discussion

We have performed an open-label feasibility study of G-CSF therapy in seven patients with FA. One participant failed to complete the trial; the remaining six participants completed the study with minimal side effects. There were no serious adverse events. HSC mobilisation was achieved by all participants in response to G-CSF treatment. Significant and sustained increases in both frataxin and mitochondrial and/or antioxidant molecules associated with the pathogenesis of FA were observed.

G-CSF is a glycoprotein with multiple biological functions. It is a well-characterised regulator of haematopoiesis and the innate immune system, regulating the survival, development, activation and mobilisation of neutrophils and haematopoietic stem and precursor cells[12]. G-CSF has been used for many years in clinical practice for the treatment of neutropenia or to stimulate the mobilisation of HSCs prior to a PB stem cell harvest. Consequently, the pharmacokinetics of G-CSF administration have been extensively studied. G-CSF has a well-established safety record and haematological profile, although has not been previously tested in people with FA. Crucially, G-CSF administered to participants of the trial, at a dose matching that given to healthy donors prior to a PB stem cell harvest, was well tolerated. One participant withdrew after the initial dose of G-CSF due to nausea; however, it was unclear whether this was related to the G-CSF administration. Of the mild adverse effects, musculoskeletal pain and headache were the only symptoms recorded, which resolved upon cessation of the therapy. Bone pain is a known common side effect of G-CSF administration, likely due to large numbers of granulocytes within the bone marrow and histamine release, which acts as a chemical mediator of inflammation and local oedema[17]. Clinical trials of G-CSF administration in other neurological conditions, such as amyotrophic lateral sclerosis and stroke, have been performed and similarly indicated good safety and tolerability of the drug[18,19].

G-CSF could have beneficial effects in FA through a range of mechanisms, including direct effects within the CNS, and the indirect consequences of its actions on bone marrow. G-CSF can cross the blood–brain barrier[20], and has multiple trophic effects on CNS cells paralleling numerous cellular functions in analogy to its role in the haematopoietic system[21]; therapeutically, it is also known to exert direct neuroprotective and neuro-regenerative effects on neural cells[22]. A conventional course of G-CSF induces the mobilisation of large numbers of CD34+ HSCs into the PB, although the resulting number of circulating CD34+ cells can be variable depending on a variety of physiological factors[23]. Impaired mobilisation of progenitor cells may be inherent to neurological disease. Pathogenic disruption of sympathetic innervation of the bone marrow, as apparent in diabetic autonomic neuropathy, has been shown to impair stem cell mobilisation, although there is thought to be only minor involvement of the autonomic nervous system in FA[24,25]. Importantly, however, we have shown that in people with FA, the haematological profile and mobilisation of HSCs in response to G-CSF administration is comparable to both healthy individuals[13] and those with other neurological conditions[18,19]. We have also provided evidence for G-CSF-induced mobilisation of CD34+ CD133+ cells, a distinct class of primitive HSCs with increased proliferation and differentiation potential[26]. Release of haematopoietic stem and progenitor cells into the peripheral circulation may have important therapeutic implications in FA. Our studies investigating bone marrow transplantation strategies in humanised FA mouse models have revealed that G-CSF can stimulate the recruitment of bone marrow cells to areas of FA-associated pathology to promote nerve cell repair[10].

We found sustained elevations in frataxin protein concentration (mean and proportional change), for up to two weeks post treatment, in both PB MNCs and platelets after the 5-day course of G-CSF. Since differences in the time course of frataxin deviations were observed, we also analysed the '$P$max' for frataxin levels to determine the maximum change in frataxin concentration over the entire study period, which was positive for all

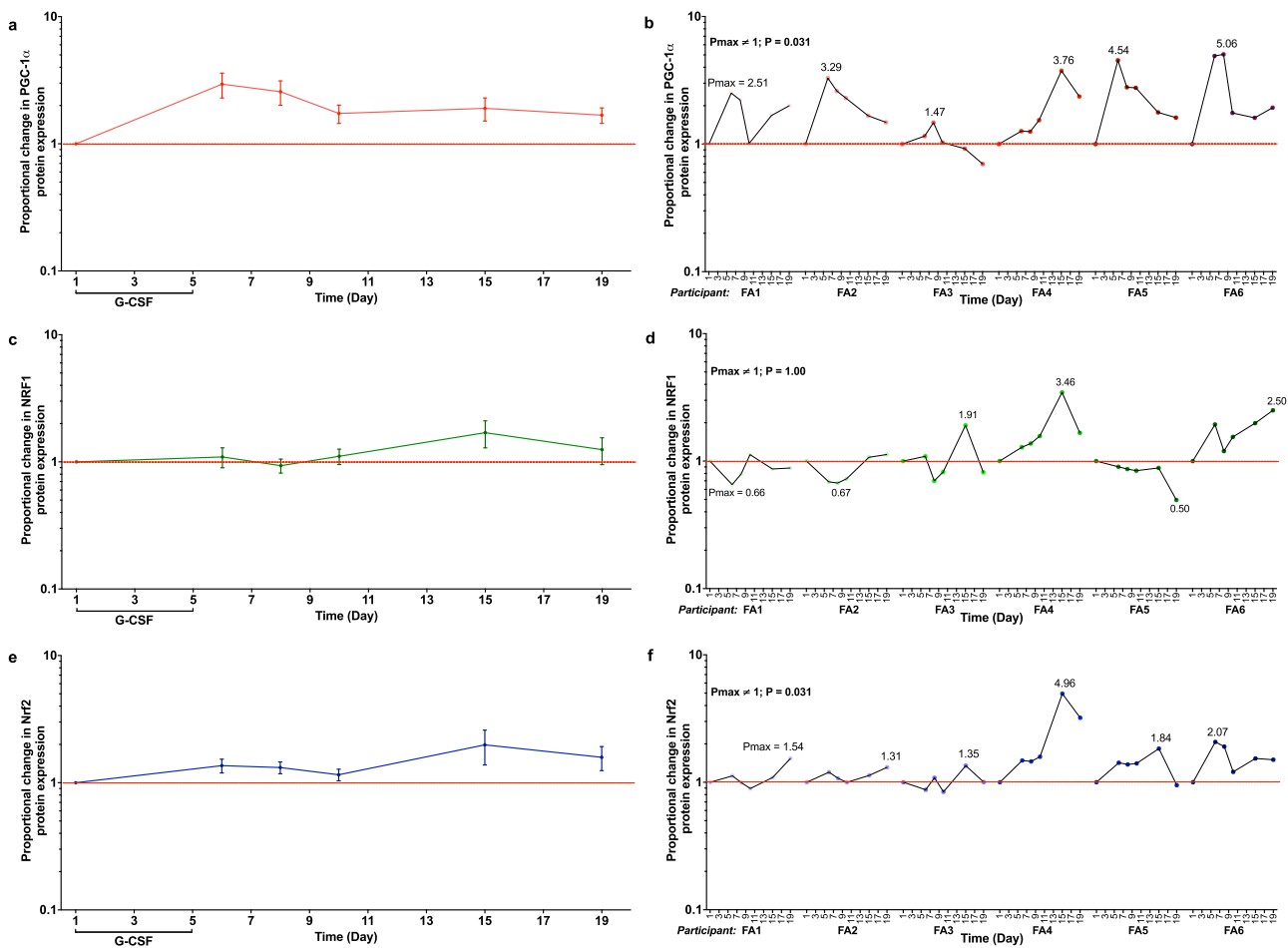

**Fig. 5 Antioxidant defences are elevated in PB MNCs in response to G-CSF administration.** The proportional changes in (**a**) PGC-1α, (**c**) NRF1 and (**e**) Nrf2 protein expression in PB MNCs, relative to baseline (day 1) levels, over the 19-day study period (graph depicts mean ± standard error; $n = 6$ biologically independent samples); data from individual participants are shown in graphs (**b**; PGC-1α), (**d**; NRF1) and (**f**; Nrf2). The sign test (two-sided) was used to compare $P$max values to a theoretical median of 1.0. Source data are provided as a Source Data file.

participants. These observations are consistent with our initial reports of G-CSF having pronounced effects on frataxin levels in the brain and spinal cord of FA mice[11]. The data also suggest that quantification of frataxin in platelets may be an attractive biomarker for potential future clinical trials relating to G-CSF; platelets are a relatively homogenous and stable cellular fraction, technically easier to isolate than MNCs and contain 86% of the total mitochondria present in the whole blood[27].

A therapy able to increase the expression of frataxin protein is an attractive treatment approach for FA; residual frataxin protein levels in FA are reduced to 5–35% of healthy individuals, while carriers of the GAA expansion having approximately 50% of normal frataxin expression are asymptomatic[28]. Baseline MNC and/or platelet frataxin concentrations reported within our study were consistent with those reported by others[7,27,29,30]. Furthermore, extrapolating data from those studies that reported frataxin levels in healthy control patients, frataxin levels in the majority of participants were brought within the predicted range for carriers of the GAA expansion following G-CSF treatment[27,30]. Further investigation into the effects of repeated G-CSF dosing on both frataxin expression and clinical measures is therefore warranted.

Oxidative damage and impaired mitochondrial function are both key determinants of cellular damage in FA. Frataxin is crucial for iron-sulphur cluster formation, which is required for the functional activity of the mitochondrial enzymes SDH and aconitase[3]. In response to G-CSF and elevations in MNC frataxin

levels, this initial study did not provide convincing evidence of subsequent increases in aconitase or SDH activity, although the mean levels of both remained above baseline for the duration of the study and the maximal change in SDH activity ($P$max) being an increase relative to baseline in five of the six participants. PGC-1α, NRF1 and Nrf2 are key orchestrators of mitochondrial biogenesis and antioxidant responses and their expression/activity is inhibited in cells from individuals with FA, leading to cell injury[14–16]. In parallel with increases in frataxin expression, G-CSF therapy-induced rapid and sustained increases in PGC-1α levels followed by a delayed but significant increase in Nrf2. This is consistent with PGC-1α's role in the regulation of Nrf2 activation[31]. NRF1, another downstream target of PGC-1α and known regulator of mitochondrial biosynthesis[14], was unchanged. Such findings suggest complex cell signalling pathways are involved in response to G-CSF treatment. Moreover, G-CSF-induced activation of PGC-1α and Nrf2 signalling may be another valuable therapeutic mechanism to explore in FA.

Pharmacological interventions to increase frataxin expression and reverse the deleterious effects of frataxin deficiency are attractive therapeutic approaches in FA. We have shown that a course of G-CSF therapy in participants with FA is safe, is associated with effective HSC mobilisation, and leads to significant elevations in frataxin together with improvements in biochemical deficits associated with FA. Nevertheless, the need for future assessment of G-CSF administration on affected tissues,

such as the heart and brain, using a range of dose levels and dosing frequencies is required. The long-term safety of sustained G-CSF administration in people with FA is also unknown. The natural rate of disease progression in FA necessitates prolonged trial periods to sufficiently detect changes in clinical measures. This study provides proof-of-principle evidence to support an efficacy study of G-CSF administration in FA, using repeated courses over a longer period.

## Methods

**Participants and study design.** We performed an open-label, single-site pilot study of Lenograstim (recombinant human G-CSF; Granocyte™) administration in seven people with FA. Participants were recruited via Neurology Clinics at North Bristol NHS Trust (Southmead Hospital, Bristol, UK). Male and female participants aged over 18 years were eligible to participate if they had a genetically confirmed diagnosis of FA (GAA-repeat expansion on both alleles of the *FXN* gene). Exclusion criteria included participation in other clinical trials within 30 days of the initial dose of G-CSF; pregnancy, breastfeeding or lactation; current or previous diagnosis of serious medical disorders or illnesses including haematologic disease (including malignancy), splenomegaly, autoimmune disease, pulmonary infiltrate, pulmonary fibrosis or haemoptysis; and clinical abnormalities on baseline blood (FBC, renal and liver function).

The study was approved by the UK Medicines and Healthcare Products Regulatory Agency (EudraCT: 2017-003084-34), the East of England – Cambridge East Research Ethics Committee (17/EE/0486) and the University of Bristol, Research Governance Research & Enterprise Development Office. Patients were enrolled in the study between 25 June 2018 to 15 October 2018. Informed written consent was provided by all participants enrolled in the study.

**Trial schedule.** Trial participants attended the Bristol Brain Centre, Southmead Hospital (Bristol, UK) for all clinical assessments and procedures. On day 1, before drug administration, baseline blood tests (FBC, renal and liver function) were completed, a baseline PB sample was collected for laboratory analysis, and clinical assessment of weight, pulse, blood pressure and temperature taken (Fig. 1). G-CSF (1.28 million units per kg) was subsequently administered by a trained physician subcutaneously, once daily, for 5 consecutive days. Further blood samples for laboratory analysis including renal and liver function tests as well as FBC were taken on days 5, 6, 8, 10, 15 and 19 (Fig. 1). Clinical examinations of pulse, blood pressure and temperature were repeated on days 5 and 19.

During the trial, all adverse events (any untoward medical occurrence, including abnormal laboratory findings, regardless of the likelihood of causal relationship to G-CSF administration) were recorded.

**Blood separation.** Whole blood, collected into EDTA Vacutainers™ (BD Bioscience), was centrifuged at 900 × g for 10 min at 4 °C to separate plasma from blood cells. MNCs were isolated from the cell fraction by density gradient centrifugation using Lymphprep™ (StemCell Technologies). Platelets were isolated by centrifugation of plasma (2500 × g for 10 min at 4 °C).

**Quantification of PB HSCs.** On days 1 and 5, peripheral whole blood samples were analysed by flow cytometry. Blood samples were incubated with antibodies anti-CD45-PE (1:5; 555483; BD Biosciences), anti-CD34-FITC (1:5; 555821; BD Biosciences) and anti-CD133-APC (1:10; 130-113-106; Miltenyi Biotec) in BD Trucount tubes (BD Biosciences) for 15 min at room temperature. Mature red blood cells were lysed with FACS lysis buffer (BD Biosciences) and samples suspended in phosphate-buffered saline. At least 85,000 events per blood sample were acquired with a BD FACSCanto flow cytometer (BD Biosciences). CD34 and CD133 positive stem cells were quantified using BD FACSDiva v8.0.1 software (BD Biosciences), based on the standardised International Society for Hematothery and Graft Engineering gating protocol[32] (Supplementary Fig. 1) and total white blood cell counts.

**Analysis of Frataxin protein.** Frataxin protein, in both PB MNCs and platelets, was measured using the Frataxin Protein Quantity Dipstick Assay Kit (Abcam) according to the manufacturer's instructions. Total protein concentrations of MNC or platelet lysates were quantified using the Pierce™ BCA Protein Assay Kit (ThermoFisher Scientific). In all, 2.5 or 10 μg of total protein from MNCs and platelets, respectively, was loaded per dipstick. Frataxin values were obtained using a standard curve generated using human recombinant frataxin protein (Abcam). Frataxin signal intensity was visualised using a Bio-Rad Universal III Bioplex imager and quantified using ImageJ software (ImageJ version 2.0.0-rc-43/1.52n [2015], NIH, Maryland, USA). Local background intensity was calculated for each separate dipstick and subtracted from the frataxin signal intensity value. Final frataxin protein values were calculated as pg of frataxin per 1 μg of total protein.

**Analysis of aconitase and succinate dehydrogenase activity.** The enzyme activities (U) of both aconitase and SDH within PB MNC homogenates were determined using the Aconitase Activity Assay Kit and Succinate Dehydrogenase Activity Colorimetric Assay Kit (MAK051 and MAK197; both Sigma-Aldrich, UK) according to the manufacturer's instruction. Absorbance was measured using a FLUOstar OPTIMA plate reader (BMG Labtech) and OPTIMA software (BMG Labtech; version 2.20R2). Homogenate total protein concentration was quantified using the Pierce™ BCA Protein Assay Kit (ThermoFisher Scientific). Enzyme activity values were calculated as activity per 1 μg of total protein.

**Analysis of NRF1, Nrf2 and PGC-1α, proteins.** Quantification of nuclear respiratory factor 1 (NRF1), nuclear factor E2-related factor 2 (Nrf2) and peroxisome proliferator-activated receptor-gamma coactivator 1 alpha (PGC-1α) was carried out using the Bio-Dot Microfiltration manifold system (Bio-Rad Laboratories). Proteins were extracted from MNCs using Extraction Buffer (ab193970; Abcam), transferred to nitrocellulose membrane using gravity filtration, and blocked using 5% bovine serum albumin, before incubation with antibodies to beta-actin (1:5000; ab8227; Abcam), NRF1 (1:10,000; ab175932; Abcam), Nrf2 (1:3,000; sc-722; Santa Cruz Biotechnology, Santa Cruz, CA) and PGC-1α (1:3,000; sc-13067; Santa Cruz Biotechnology). Immunoreactivity was detected using a horseradish peroxidase–conjugated goat anti-rabbit IgG (1:3,000; ab6721; Abcam) secondary antibody. Protein expression was visualised using ECL Prime Western Blotting Detection reagent (GE Healthcare) in conjunction with a Bio-Rad Universal III Bioplex imager. Densitometric analysis of protein expression was performed using Image Lab software (Image Lab™ version 6.0.1 [2017], Bio-Rad laboratories). Relative expression of NRF1, Nrf2 and PGC-1α was calculated after normalisation to beta-actin levels within the same sample.

**Sample size.** No sample size calculations were performed. This was a pilot study of seven patients prior to larger phase 2/3 studies. Sample sizes were designed to characterise the outcome measures (including standard deviation required for power calculations in future studies) and based on the incidence of disease in the population, willingness of patients to consider recruitment, adherence rates and time required for data collection.

**Statistics and reproducibility.** Analyses were performed using GraphPad Prism (GraphPad Prism version 8.4.1 for macOS (2020), GraphPad Software, USA). All measurements were taken from independent samples/biological replicates provided by each of the study participants at multiple time points. For each measure of protein concentration or enzyme activity, the maximum absolute deviation from the baseline value was determined for each of the six participants ($P$max). We present these individual data graphically, with each of the measurements taken from an individual being divided by that individual's baseline level, and presented on a log scale to facilitate the distinction between maximum deviations which are greater than or reduced from the baseline value. The sign test (two-sided) quantified evidence against the null hypothesis that an equal number of participants have $P$max values that are increased or decreased compared to their baseline value, i.e. changes are due to random variation or measurement error rather than a systematic effect of Lenograstim.

**Reporting summary.** Further information on experimental design is available in the Nature Research Reporting Summary linked to this paper.

## Data availability

The authors declare that all data supporting the findings of this study are available within the paper and its Supplementary information. Raw data used to calculate deviations from baseline values (Figs. 3–5) are included in the Source data file. The study protocol is provided with this paper as Supplementary methods. Source Data are provided with this paper.

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

## Acknowledgements

We thank Ataxia UK for their assistance with recruiting participants and the North Bristol NHS Trust Research and Innovation for hosting the trial. The authors thank the participants, their families and carers for their generous time and commitment to this study. We thank Nigel Noel and Paul Virgo, at The Department of Immunology and Immunogenetics, North Bristol NHS Trust (Southmead Hospital, Bristol, UK), for their assistance with flow cytometry data acquisition. The study was funded by Ataxia UK; the Friedreich's Ataxia Research Alliance (FARA); The Burden Institute; the Elizabeth Blackwell Institute; the Wellcome Trust [204813/Z/16/Z - ISSF3].

## Author contributions

A.W., K.C.K. and N.J.S. were involved in the study conception and design. A.W. recruited participants. A.W. and S.B. performed clinical assessments and took blood samples. K.C.K., A.G., K.H. and J.R. performed laboratory tests. K.C.K., A.G., C.M.R. and C.M. performed data analysis. A.W., K.C.K. and C.M. interpreted the data. K.C.K., A.W., C.M.R., N.J.S. and C.M. helped write the report.

## Competing interests

The authors declare no competing interests.
