## [Peer review file · Nature Communications]

An open-label pilot study of recombinant granulocyte-colony stimulating factor in Friedreich's ataxiaReviewers' comments:

Reviewer #1 (Remarks to the Author):

This is an interesting manuscript describing a phase I trial of gCSF in Friedreich ataxia. I think the study is well designed, and appropriately no direct clinical information is reported (the study is underpowered for such aspects). There are however, a few aspects that could be included that would extend its meaning.

1. There needs to be a full description of subjects—levels of severity, age of onset, age GAA repeat length. Do any of the results correlate with these features?
2. In frataxin measurements, it is extremely disappointing that the data do not show when levels returned to baseline. The same comment would be true for PGC1a/NRF2/mitochondrial enzymes
3. It is also disappointing that two other tissues were not examined –muscle (an affected tissue) and RBC (made from a different splice variant).
4. It would be very helpful if the levels of response were compared to control or carrier levels to provide a perspective.

Reviewer #2 (Remarks to the Author):

Kemp and colleagues report an open-label pilot study using G-CSF in Friedreich's ataxia (FA). They have tested in seven subjects with FA the effect on hematological profile and frataxin concentration after 5 days of levograstim as well as the safety of the drug. The results are similar of the one found in other diseases and in healthy subject, with an increase of the hematological parameters. They show in addition an increase of the frataxin concentration. The side effects are as expected with the drug.

There are some comments.

The sign test is a test used to compare pairs of observation. What were the groups compared using this test (page 6 line 175)? What was the test used to compare the Pmax values to the theoretical median of 1.0. It is written page 6 line 178 that it is the sign test. However this is a paired test not a test to compare to a reference value. In addition if sign tests are used, they do not compare the means. Thus, medians and interquartile ranges should be reported instead of means and standard deviations (in the results section, in Table and figures).

On participant failed. What were his clinical characteristics?

A description of the baseline characteristics of the participants is given (Table 1). What was the clinical severity of the subjects at baseline? What was the number of GAA repeat length? Even if the drug was given for a short duration, what was the clinical evolution after treatment?

Figure 3, 4 and 5 are difficult to read. The font is too small. In addition, the way individual proportional changes are plotted is confusing.

Reviewer #3 (Remarks to the Author):

Important baseline for possible future larger controlled trial of GCSF in FA patients.

An open-label pilot study of recombinant granulocyte-colony stimulating factor in Friedreich's ataxia

Kevin C. Kemp, Anastasia Georgievskaya, Kelly Hares, Juliana Redondo, Steven Bailey, Claire M. Rice, Neil J. Scolding, Chris Metcalfe and Alastair Wilkins

We thank both the reviewers for their helpful and positive comments pertaining to this paper. We believe we have fully responded to them all and have submitted our revised manuscript in line with these comments. Changes within the manuscript are highlighted in red text.

General comments to reviewers

We apologize for a labelling error to Fig.4b. Pmax values for changes in aconitase activity were positive in four (rather than five) of the six participants. Both the figure and text (lines 241 & 330) have been updated in line with these changes.

Reviewer #1 (Remarks to the Author):

Comment 1. This is an interesting manuscript describing a phase I trial of GCSF in Friedreich ataxia. I think the study is well designed, and appropriately no direct clinical information is reported (the study is underpowered for such aspects). There are however, a few aspects that could be included that would extend its meaning. There needs to be a full description of subjects — levels of severity, age of onset, age GAA repeat length. Do any of the results correlate with these features?

Answer: We thank the reviewer for their suggestions. Table 1 has now been adjusted to present individual patient data (rather than summary statistics) to include both data on age at onset and clinical phenotype (including disability staging and speech scores). All participants had a genetically confirmed diagnosis of FA (GAA-repeat expansion on both alleles of the FXN gene). GAA repeat lengths were not recorded in this study in keeping with the trial protocol.

We agree with the reviewer that it would be interesting to correlate our results with patient demographics. However, the current study – a pilot study designed principally to explore safety and tolerability – was not powered to determine whether frataxin levels (or other biochemical markers) correlate with clinical features, making it difficult to interpret any null finding.

Comment 2. In frataxin measurements, it is extremely disappointing that the data do not show when levels returned to baseline. The same comment would be true for PGC1a/NRF2/mitochondrial enzymes.

Answer: This is an important consideration. For this study, our main outcome objectives were to determine whether G-CSF increased frataxin/antioxidant expression, and the time scale/level of peak increases (both critical for monitoring levels in future clinical trials). In line with this, we have recently developed a more sensitive method for detecting dipstick signal intensity. We plan to use this detection

method for our future studies. Dipstick MNC and platelet frataxin signal intensity levels have therefore been reanalyzed in light of these changes and the methods section updated. Mean baseline frataxin levels were unchanged upon reanalysis and Pmax values remain positive for all six participants. We agree that it would have been scientifically interesting to determine whether all levels returned to baseline (although some participants did return to baseline; see Figs. 3 and 5). However, the protocol/ethical approval for the trial necessitated final blood sampling at day 19 and we were unable to collect later blood samples. Future trials being planned will address this important issue.

Comment 3. It is also disappointing that two other tissues were not examined – muscle (an affected tissue) and RBC (made from a different splice variant).

Answer: We agree it would be interesting to monitor changes in alternative frataxin splice variants, such as isoform E within RBCs, however mature erythrocytes lack both nuclei/mitochondria, and have a half-life of approximately 100 days, which make detecting changes in frataxin protein expression over short periods of time extremely difficult [Guo et al. *Anal Chem.* 2018 Feb 6; 90(3): 2216–2223].

Due to its invasive nature and high sampling variance, authors believed taking repeated muscle biopsies (six biopsies taken over 19 days - to match blood samples) was not justified for this pilot study.

Comment 4. It would be very helpful if the levels of response were compared to control or carrier levels to provide a perspective.

Answer: We agree this is an important point and we have now included this comparison. It is well characterized that frataxin expression levels in patients with FA correlate with both the GAA-repeat expansion length and clinical severity - residual frataxin protein levels in FA are reduced to 5 - 35% of healthy individuals, while carriers of the GAA expansion having approximately 50% of normal frataxin expression, are asymptomatic. Baseline frataxin MNC and/or platelet concentrations reported within our study were found to be consistent with those reported by others^{1,2,3,4}. Furthermore, extrapolating data from those studies that reported frataxin levels in healthy control patients, frataxin levels in the majority (four out of six) of our participants were brought within the predicted range for carriers of the GAA expansion following G-CSF treatment^{1,2}. Comparisons of the frataxin levels detected within this study to those reported in FA, carrier and control patients have been added to the discussion section.

1. Guo L, et al. Liquid Chromatography-High Resolution Mass Spectrometry Analysis of Platelet Frataxin as a Protein Biomarker for the Rare Disease Friedreich's Ataxia. *Anal Chem* **90**, 2216-2223 (2018).
2. Steinkellner H, Scheiber-Mojdehkar B, Goldenberg H, Sturm B. A high throughput electrochemiluminescence assay for the quantification of frataxin protein levels. *Anal Chim Acta* **659**, 129-132 (2010).
3. Nachbauer W, et al. Effects of erythropoietin on frataxin levels and mitochondrial function in Friedreich ataxia--a dose-response trial. *Cerebellum* **10**, 763-769 (2011).
4. Mariotti C, et al. Erythropoietin in Friedreich ataxia: no effect on frataxin in a randomized controlled trial. *Mov Disord* **27**, 446-449 (2012).

Reviewer #2 (Remarks to the Author):

Comment 1. The sign test is a test used to compare pairs of observation. What were the groups compared using this test (page 6 line 175)? What was the test used to compare the Pmax values to the theoretical median of 1.0. It is written page 6 line 178 that it is the sign test. However this is a paired test not a test to compare to a reference value. In addition if sign tests are used, they do not compare the means. Thus, medians and interquartile ranges should be reported instead of means and standard deviations (in the results section, in Table and figures).

Answer: The sign test is based on the binomial probability distribution, and can be considered to test the null hypothesis of "No systematic change in level following intervention". The p-value is calculated as the $\text{prob}(X \geq x | p=0.5)$ using the binomial distribution, i.e. in the absence of an intervention effect, random measurement error and individual variation will lead to an equal number of participants with positive and negative change. This does not rely on a particular summary statistic, and for the most part we also present the individual data which is arguably the most useful in interpreting the sign-test p-value. We have added in further clarification at page 7 line 194.

Comment 2. On participant failed. What were his clinical characteristics?

Answer: We have expanded Table 1 to include baseline clinical characteristics. The participant that failed the study is now identified within Table 1.

Comment 3. A description of the baseline characteristics of the participants is given (Table 1). What was the clinical severity of the subjects at baseline? What was the number of GAA repeat length? Even if the drug was given for a short duration, what was the clinical evolution after treatment?

Answer: We thank the reviewer for their suggestions. Table 1 has now been adjusted to present individual patient data (rather than summary statistics) to include both data on age at onset and baseline clinical severity (including disability staging and speech scores). All participants had a genetically confirmed diagnosis of FA (GAA-repeat expansion on both alleles of the FXN gene). GAA repeat lengths and detailed ataxia rating scales were not recorded in this study in keeping with the trial protocol, as the current pilot study was not powered to report the effects of treatment on clinical features.

Comment 4. Figure 3, 4 and 5 are difficult to read. The font is too small. In addition, the way individual proportional changes are plotted is confusing.

Answer: We thank the reviewer for these helpful comments. All figures have been changed accordingly to maximise font size. Additional labelling has also been added to plots displaying individual proportional changes to make them clearer.

Reviewer #3 (Remarks to the Author):

Comment 1. Important baseline for possible future larger controlled trial of GCSF in FA patients.

Answer: We thank the reviewer for their positive comment.

REVIEWERS' COMMENTS:

Reviewer #1 (Remarks to the Author):

The authors have done a good job of responding.

There is one further item perhaps. In my opinion, the work would be strengthened by a paragraph in the end acknowledging the limitations of immediate extrapolation to people--lack of evidence in affected tissue, full dose range not yet defined, longterm side effects in FRDA, no evidence yet of effects in unaffected tissue. While these things are beyond what one might expect in a phase I study, it is important for the community to remember that these issues are not yet resolved.

Reviewer #2 (Remarks to the Author):

The authors answered to my comments. It is a pity that GAA repeat length are not available. I have no additional comments.

NCOMMS-19-38635B: response to reviewer's comments

An open-label pilot study of recombinant granulocyte-colony stimulating factor in Friedreich's ataxia

Kevin C. Kemp, Anastasia Georgievskaya, Kelly Hares, Juliana Redondo, Steven Bailey, Claire M. Rice, Neil J. Scolding, Chris Metcalfe and Alastair Wilkins

We thank again both the editor and reviewers for their helpful and positive comments pertaining to this paper. We believe we have fully responded to them all and have submitted our revised manuscript in line with these comments. Changes within the manuscript are highlighted using tracked changes.

Reviewer #1 (Remarks to the Author):

Comment 1. The authors have done a good job of responding. There is one further item perhaps. In my opinion, the work would be strengthened by a paragraph in the end acknowledging the limitations of immediate extrapolation to people--lack of evidence in affected tissue, full dose range not yet defined, longterm side effects in FRDA, no evidence yet of effects in unaffected tissue. While these things are beyond what one might expect in a phase I study, it is important for the community to remember that these issues are not yet resolved.

Answer: We thank the reviewer for this suggestion. A paragraph has been added to the end of the discussion acknowledging the limitations of this pilot study.